# Bottleneck Effect Explained by Le Bail Refinements: Structure Transformation of Mg-CUK-1 by Confining H_2_O Molecules

**DOI:** 10.3390/ma13081840

**Published:** 2020-04-14

**Authors:** Elí Sánchez-González, J. Gabriel Flores, Julio C. Flores-Reyes, Ivette Morales-Salazar, Roberto E. Blanco-Carapia, Mónica A. Rincón-Guevara, Alejandro Islas-Jácome, Eduardo González-Zamora, Julia Aguilar-Pliego, Ilich A. Ibarra

**Affiliations:** 1Laboratorio de Fisicoquímica y Reactividad de Superficies, Instituto de Investigaciones en Materiales, Universidad Nacional Autónoma de México, Circuito Exterior S/N, Ciudad Universitaria, Coyoacán, C.P. 04510 Ciudad de México, Mexico; sanchez.gonzalez.eli@gmail.com (E.S.-G.); gabriel_flores_aguilar@hotmail.com (J.G.F.); 2Institute for Integrated Cell-Material Sciences (WPI-iCeMS), Kyoto University, Yoshida, Sakyo-ku, Kyoto 606-8501, Japan; 3Departamento de Química Aplicada, Universidad Autónoma Metropolitana-Azcapotzalco, San Pablo 180, Col. Reynosa-Tamaulipas, Azcapotzalco, C.P. 02200 Ciudad de México, Mexico; 4Instituto de Catálisis y Petroleoquímica, ICP-CSIC, C/ Marie Curie, 2, C.P. 28049 Madrid, Spain; 5Departamento de Química, Universidad Autónoma Metropolitana-Iztapalapa, San Rafael Atlixco 186, Col. Vicentina, Iztapalapa, C.P. 09340 Ciudad de México, Mexico; flores.reyes.jc@gmail.com (J.C.F.-R.); ivette649_tatu@hotmail.com (I.M.-S.); edreyblanco@gmail.com (R.E.B.-C.); aij@xanum.uam.mx (A.I.-J.); 6Departamento de Biotecnología, Universidad Autónoma Metropolitana-Iztapalapa, San Rafael Atlixco 186, Col. Vicentina, Iztapalapa, C.P. 09340 Ciudad de México, Mexico; monicarinconguevara@gmail.com

**Keywords:** MOFs, structure transformation, PXRD Le Bail refinements

## Abstract

The structure transformation of Mg-CUK-1 due to the confinement of H_2_O molecules was investigated. Powder X-ray diffraction (PXRD) patterns were collected at different H_2_O loadings and the cell parameters of the H_2_O-loaded Mg-CUK-1 material were determined by the Le Bail strategy refinements. A bottleneck effect was observed when one hydrogen-bonded H_2_O molecule per unit cell (18% relative humidity (RH)) was confined within Mg-CUK-1, confirming the increase in the CO_2_ capture for Mg-CUK-1.

## 1. Introduction

Global warming is one of the greatest risks to human civilization. In particular, the growing levels of anthropogenic carbon dioxide (CO_2_) emissions from fossil fuel combustion [1] directly influence our environment, triggering the continuous rise of temperatures across the planet. Only in 2017, worldwide CO_2_ emissions from fossil fuel combustion augmented by approximately 2% compared with the 2015–2016 period [2]. Currently, governments are working together on a worldwide basis to encourage the development of new technologies for a more efficient and effective CO_2_ capture [3]. Porous metal–organic frameworks (MOFs) or porous coordination polymers (PCPs) are among the most promising candidates for CO_2_ capture because their carbon dioxide sorption properties can be more broadly tuned than classical mesoporous materials (e.g., zeolites) [4]. Current synthetic approaches to further increase sorption selectivity towards carbon monoxide include the incorporation of open metal sites that can enhance molecular sorption, and by functionalizing the organic linker with the Lewis basic groups (e.g., amines and alcohols) [5]. Very recently, thorough investigations were also made to enhance the CO_2_ capture of MOFs by using the synergistic effects caused by pre-confining small amounts of polar molecules in their pores [6,7,8]. In this regard, we previously showed that the confinement of small amounts of H_2_O in a series of MOFs (functionalized with hydroxo functional groups, *μ*_2_-OH) steadily resulted in improved CO_2_ capture properties [9]. The confined H_2_O molecules are well-ordered in the pore-structure of these *μ*_2_-OH functionalized MOFs, working as preferential adsorption sites for the subsequent CO_2_ molecules [10]. In this communication, we describe the structure transformation of Mg-CUK-1 (CUK for Cambridge University–KRICT, see Appendix A) [11] due to the confined H_2_O molecules within its pores, which previously demonstrated enhanced CO_2_ capture properties [12,13].

## 2. Materials and Methods 

### 2.1. Material Synthesis

Mg-CUK-1 = [Mg_3_(OH)_2_(2,4-PDC)_2_, 2,4-PCD = 2,4-pyridinecarboxylate] was synthesized following the previously reported procedure in [11]: 2,4-Pyridinedicarboxlic acid (170 mg, 1.0 mmol) and KOH (2.0 _M_, 2.0 cm^3^) in H_2_O were added to a stirred solution of Mg(NO_3_)_2_, (380 mg, 1.5 mmol) in H_2_O (3 cm^3^) to give a viscous, opaque, slurry mixture. The reaction mixture was placed inside a Teflon-lined Easy-Prep vessel and heated at 472 K for 35 min in a MARS microwave (CEM Corp.). The reaction temperature was monitored using a fiber-optic sensor. After cooling down to room temperature (30 min), the crystalline solid was purified by short (3 × 20 s) cycles of sonication in fresh H_2_O, followed by the decanting of the slurry supernatant. Large, colorless prismatic crystals were isolated (average yield: 124 mg). TGA and PXRD were carried out and confirmed the nature of the synthesized material and its purity (see Appendix A), and the estimated BET surface area of the activated Mg-CUK-1 (100 °C at 1 × 10^−4^ bar and 2 h).

### 2.2. Methods

Adsorption Isotherms for CO_2_. Ultra-pure grade (99.9995%) CO_2_ gas was purchased from PRAXAIR. CO_2_ adsorption isotherms at 196 K and up to 1 bar were carried out on a Belsorp mini II analyzer under high vacuum.

Water-loading within Mg-CUK-1. H_2_O vapor loadings were performed by a dynamic method, using air gas as the carrier gas, and by using a DVS Advantage 1 instrument from Surface Measurement Systems (mass sensitivity: 0.1 mg, relative humidity (RH), accuracy: 0.5% RH, vapor pressure accuracy: 0.7% P/P_0_). Mg-CUK-1 samples were activated at 100 °C for 1 h under flowing dry N_2_. The water contents within Mg-CUK-1 analyzed were 25% RH, 22% RH, 20% RH and 18% RH.

Powder X-ray diffraction patterns were collected on a Rigaku Diffractometer, Ultima IV, with a Cu-Kα1 radiation (λ = 1.5406 Å) using a nickel filter. These were obtained from 5° to 50° in 2θ, with 0.02° steps at a 0.08° min^−1^ scan rate. Profile refinements were performed based on the previously reported Mg-CUK-1-hydrated structure data using the FullProf program (structure NUDLIJ from CCDC database) [14,15]. 

## 3. Results

Mg-CUK-1 is assembled from the coordination of Mg(II) metal centers and 2,4-pyridinedicarboxylate ligand. Mg-CUK-1 crystallizes in the space group P2_1_/*n* and it is constructed around trinuclear [Mg_3_(*μ*_3_-OH)] building blocks (see Appendix A, inset) [11]. Each Mg(II) metal center shows an octahedral coordination environment and links into infinite chains of edge- and vertex-sharing Mg_3_OH triangles. Mg-CUK-1 shows a 3-D framework structure with diamond-shaped pore dimensions of approximately 8.1 × 10.6 Å (see Appendix A). The estimated BET area (0.005 < P/P0 < 0.15) was equal to 604 m^2^ g^−1^, with a corresponding pore volume of 0.22 cm^3^ g^−1^.

The water-loading dependence of the porosity of Mg-CUK-1 was analyzed based on the reported [12,13] water adsorption isotherm data. For that purpose, the experimental PXRD patterns were collected at different H_2_O loadings (see Appendix A, water-loading within Mg-CUK-1, Appendix A) and the cell parameters of the H_2_O-loaded material (Mg-CUK-1) were determined by the Le Bail methodology (FullProf program; see Appendix A, PXRD profile refinement of Mg-CUK-1, Appendix A) [14,15]. The so-obtained evolution of the cell parameters corroborated the soft crystal properties of Mg-CUK-1: the *b*-axis increases with the water content from 12.334 to 13.435 Å (see Appendix A). Such a change occurred from the anhydrous form to the eight H_2_O/UC-loaded versions (Appendix A). We indeed observed a dramatic reduction in the accessible space when the H_2_O concentration increased (see Table 1 and Appendix A). 

The reduction in the minimum channel length is considerably more drastic in the *b* direction, which can be associated with the position of the hydroxo groups (*μ*_2_-OH) that are only present in the *b* direction of the channel. The minimum channel length in the *b* direction is reduced from 6.62 Å, in the empty material, to approximately 2.56 Å, with eight water molecules per unit cell, corresponding to more than a 50% reduction in the size of the channel diameter (Table 1). These eight water molecules correspond to one H_2_O molecule per hydroxo group and, at this point, the inclusion of CO_2_ is anticipated to not be possible. Since the kinetic diameter of CO_2_ is 3.3 Å (Figure 1), this essentially inhibits the direct passage of the CO_2_ molecules through the Mg-CUK-1 network, and therefore results in zero adsorption of CO_2_.

## 4. Discussion

Interestingly, four and two H_2_O molecules interacting via hydrogen bonding to Mg-CUK-1 (molecules per unit cell) lead to a window reduction of 4.68 and 4.43 Å, respectively (Table 1). However, the minimum distance between two adjacent water molecules is 3.3 and 6.7 Å for Mg-CUK-1 with four and two water molecules, respectively. According to this, the path for the diffusion of CO_2_ is restrained (Figure 1) and a low CO_2_ adsorption is expected. Following this trend, an increase in the distance between the water molecules within the pore can result in a favorable space for CO_2_ to interact with the H_2_O molecules. When there is one hydrogen-bonded H_2_O molecule per unit cell, the pore-window is reduced to approximately 4.43 Å and the distance between two water molecules increases to 20.5 Å (Table 1). This can provide enough free space for the CO_2_ molecules to diffuse within the pore-window and, as we previously reported, a “bottleneck effect’’ occurs [16,17]. This effect is expected to accommodate the CO_2_ molecules more efficiently, by partially obstructing the pore-windows of Mg-CUK-1. Indeed, this particular configuration (one hydrogen-bonded H_2_O molecule per unit cell) corresponds to 18% of relative humidity (RH), according to the water adsorption isotherm [12,13]. Remarkably, Mg-CUK-1 revealed a 1.8-fold increase in CO_2_ capture from 4.6 wt% to 8.5 wt% in the presence of 18% RH [12] (See Appendix A).

## 5. Conclusions

The structure transformation of Mg-CUK-1 by the incorporation (confinement) of different amounts of water molecules was successfully demonstrated by PXRD (Le Bail methodology refinements). The confinement of one hydrogen-bonded H_2_O molecule per unit cell (18% RH) produces a bottleneck effect with adequate pore dimensions (4.43 Å) for the proper diffusion of CO_2_ molecules. Such a structure transformation corroborates a 1.8-fold increase in CO_2_ capture from 4.6 wt% to 8.5 wt%, as previously reported for Mg-CUK-1 [12].

## Figures and Tables

**Figure 1 materials-13-01840-f001:**
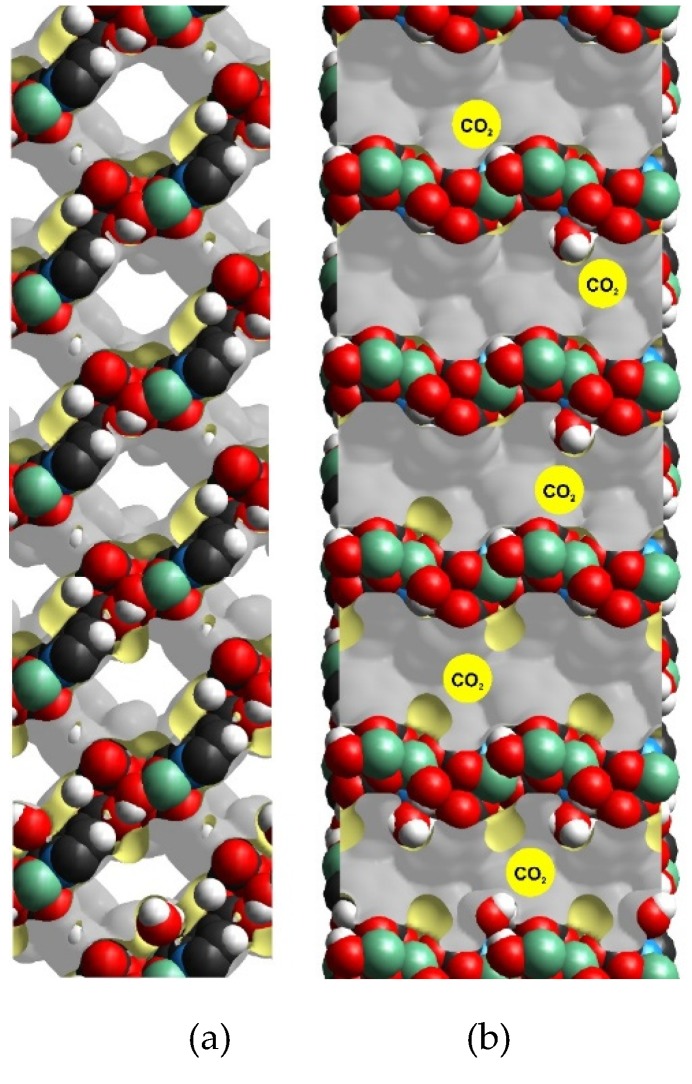
Crystal structures of Mg-CUK-1 with different H_2_O molecule loadings, from top to bottom, none, 1, 2, 4 and 8 water molecules per unit cell. (**a**) View through the *a*-axis showing the hydrogen-bonded H_2_O molecules to the hydroxo functional groups (H_2_O···OH-*μ*_3_), and (**b**) side view of the channel (through the *c*-axis) and accessible surface, yellow circles represent CO_2_ kinetic diameter.

**Table 1 materials-13-01840-t001:** Mg-CUK-1 one-dimensional channel dimensions at different H_2_O molecule loadings. The window dimensions l_b_ and l_c_ (Å) were estimated using a 0.002 au isosurface with CrystalExplorer [14,15].

Material	H_2_O/UC	l_b_ (Å)	l_c_ (Å)	O_H2O-_O_H2O_ (Å)	Pore
Mg-CUK-1	0	6.62	6.51	-	Accessible
Mg-CUK-1···1H_2_O	1	4.43	6.52	20.5	Accessible
Mg-CUK-1···2H_2_O	2	4.43	6.52	6.5 or 9.5	Restrained
Mg-CUK-1···4H_2_O	4	4.68	6.61	4.6 or 6.5	Restrained
Mg-CUK-1···8H_2_O	8	2.56	6.61	4.6	Non-accessible

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
