# Peer review of "Bottleneck Effect Explained by Le Bail Refinements: Structure Transformation of Mg-CUK-1 by Confining H2O Molecules"

_materials, 2020, doi:10.3390/ma13081840_

Round 1
Reviewer 1 Report
Review of the manuscript “Bottleneck effect explained by Le Bail refinements: Structure transformation of Mg-CUK-1 by confining H2O molecules”, by Elí Sánchez-González, J. Gabriel Flores, Ivette Morales-Salazar, Roberto E. Blanco-Carapia, Julio C. Flores-Reyes, Mónica A. Rincón-Guevara, Alejandro Islas-Jácome, Eduardo González-Zamora *, Julia Aguilar-Pliego *, Ilich A. Ibarra *, submitted to Materials.
This Communication reports a short study of sorption of water vapor on magnesium metal-organic framework (MOF) Mg-CUK-1 and determination of the resultant change in its crystalline lattice. This structural change is used to explain the change in adsorption of carbon dioxide CO2 onto Mg-CUK-1 with pre-adsorbed water.
This manuscript contains the potentially interesting data on tuning structural parameters of the lattice of this MOF by sorption of the variable amount of water vapor. However, this Communication has a critical shortage of data and information. Namely, quite many items in its “Materials and Methods” and “Results” sections are entirely missing, very poorly edited, or poorly presented. The Materials and Methods section is written particularly poorly, and has apparently not been proof-checked by authors before submission. I suggest to reject this manuscript, but encourage authors to resubmit after they implement many major changes and additions which are listed below, and have someone proof-read the manuscript before next submission. The English language of this manuscript is mainly satisfactory.
Major deficiencies
1) Page 2, lines 57-59, section “2.1. Material Synthesis”: “Mg-CUK-1 was synthesized following the previously reported procedure [11], Mg-CUK-1 = [Mg3(OH)2(2,4-PDC)2, 2,4-PCD = 2,4-pyridinecarboxylate], was synthesized following the previously reported procedure [1]:…”.
There is a duplicate statement in this sentence. Has this sentence been written by “copy-pasting” the text twice and not reading? And is this description from the reference [1] or [11]?
2) Page 2, lines 66-67: “(see Figure S2 and Figure S3, SM).”
The Supplementary Materials (SM) section is entirely absent, or at least has not been provided to this reviewer by the electronic file system of the journal.
3) Page 2, lines 67-68: “The estimated BET surface area of activated Mg-CUK-1 (100 °C at 1x10-4 bar and 2 h).”
So, what is the numeric value of the BET surface area of this material? Is this another not edited copy-pasted write-up?
4) Page 2, lines 76-77: “The water contents, within Mg-CUK-1, analyzed were 25% RH, 22% RH, 20% RH and 18% RH.”
The RH is Relative Humidity and it cannot be “within Mg-CUK-1”. The RH is defined in the gas phase in the chamber of the experimental apparatus and not inside the porous material.
5) Page 2, line 89: “at different H2O loadings (see SM, water-loading within Mg-CUK-1)”.
Again, the Supplementary Materials (SM) are absent in this submission.
6) Page 3, line 94: “anhydrous form to 8 H2O/UC loaded version (Table S1).”
The authors apparently refer to Table 1 on the same page, which is mistakenly labeled as Table S1 in the missing Supplementary Materials (SM) section.
7) Page 4, line 140: “Supplementary Materials”.
They are not available in this manuscript submitted for review.
Minor errors and typos:
1) Page 1, line 43: “selectivity towards carbon monoxide include…”.
Do authors mean “carbon dioxide” CO2?
2) Page 3, line 107-109: “Since the kinetic diameter of CO2 is 3.3 Å (Figure 1), essentially inhibits the direct passage of the CO2 molecules through the Mg-CUK-1 network and therefore resulting in zero adsorption of CO2.”
Please edit this sentence heavily and bring it to compliance with English grammar.
Author Response
Review 1
Unfortunately, there was an uploading mistake with the Supplementary Materials (MS) in this was not visible for reviewing. However, now we show in this file the requested data and the SM file for the observations 2,5,6 and 7. We have also made sure to send the SM file with these corrections.
Major deficiencies
1) Page 2, lines 57-59, section “2.1. Material Synthesis”: “Mg- CUK-1 was synthesized following the previously reported procedure [11], Mg-CUK-1 = [Mg3(OH)2(2,4-PDC)2, 2,4-PCD = 2,4-pyridinecarboxylate], was synthesized following the previously reported procedure [1]:…”.There is a duplicate statement in this sentence. Has this sentence been written by “copy-pasting” the text twice and not reading? And is this description from the reference [1] or [11]?
We have certainly made a mistake regarding the reference, it was written as [1] but the correct reference was [11] (Angew. Chem. Int. Ed. 2015, 54, 5394–5398). After the last modifications, the actual reference number is [16].
2) Page 2, lines 66-67: “(see Figure S2 and Figure S3, SM).” The Supplementary Materials (SM) section is entirely absent, or at least has not been provided to this reviewer by the electronic file system of the journal.
Both figures have been uploaded to the PDF file attached to this answer, as well as the SM.
3) Page 2, lines 67-68: “The estimated BET surface area of activated Mg-CUK-1 (100 °C at 1x10-4 bar and 2 h).” So, what is the numeric value of the BET surface area of this material? Is this another not edited copy-pasted write-up?
The numeric value of the estimated BET is in the “Results” part, in the first paragraph. “The estimated BET area (0.005 < P/P0 < 0.15) was equal to 604 m2 g-1 with a corresponding pore volume of 0.22 cm3 g-1.”
4) Page 2, lines 76-77: “The water contents, within Mg-CUK-1, analyzed were 25% RH, 22% RH, 20% RH and 18% RH.” The RH is Relative Humidity and it cannot be “within Mg- CUK-1”. The RH is defined in the gas phase in the chamber of the experimental apparatus and not inside the porous material.
We appreciate this valuable observation very much. We have changed the way we express the amount of H2O molecules in the material “Mg-CUK-1 at different relative humidity conditions (25% RH, 22% RH, 20% RH and 18% RH) was analyzed”. However, the way to measure the amount of water vapor in the DVS instrument is by relative humidity (RH).
5) Page 2, line 89: “at different H2O loadings (see SM, water loading within Mg-CUK-1)”. Again, the Supplementary Materials (SM) are absent in this submission.
Figures S4, S5, S6, and S7, corresponding to 25, 22, 20 and 18% of RH respectively, have been uploaded to the PDF file attached to this answer, as well as the SM.
6) Page 3, line 94: “anhydrous form to 8 H2O/UC loaded version (Table S1).” The authors apparently refer to Table 1 on the same page, which is mistakenly labeled as Table S1 in the missing Supplementary Materials (SM) section.
Table S1. Refinement parameters of the Mg-CUK-1 structure at different water loadings.
Formula |
Mg3(OH)2(C7H3O4N)2·XH2O |
||||
X = |
2 |
1 |
0.5 |
0.25 |
0 |
H2O per UC |
8 |
4 |
2 |
1 |
- |
Crystal system |
monoclinic |
monoclinic |
monoclinic |
monoclinic |
monoclinic |
Space group |
P21/n |
P21/n |
P21/n |
P21/n |
P21/n |
a (Å) |
10.9329 |
10.9830 |
11.0034 |
11.0135 |
11.0237 |
b (Å) |
13.4350 |
12.8277 |
12.5809 |
12.4574 |
12.3340 |
c (Å) |
17.5753 |
17.9688 |
18.1304 |
18.2116 |
18.293 |
β (°) |
104.351 |
103.305 |
102.893 |
102.690 |
102.488 |
V (Å3) |
2500.98 |
2463.61 |
2446.56 |
2437.59 |
2428.39 |
Rwp (%) |
57.0 |
50.1 |
44.2 |
52.6 |
- |
R (%) |
45.1 |
36.1 |
32.1 |
38.7 |
- |
χ2 |
51.4 |
89.1 |
72.7 |
74.7 |
- |
7) Page 4, line 140: “Supplementary Materials”. They are not available in this manuscript submitted for review.
We will make sure to send the SM file with these corrections.
Minor errors and typos
1) Page 1, line 43: “selectivity towards carbon monoxide include…”. Do authors mean “carbon dioxide” CO2?
This is indeed a mistake and now it has been modified in the main manuscript. “…increase sorption selectivity towards carbon dioxide include the incorporation…”
2) Page 3, line 107-109: “Since the kinetic diameter of CO2 is 3.3 Å (Figure 1), essentially inhibits the direct passage of the CO2 molecules through the Mg-CUK-1 network and therefore resulting in zero adsorption of CO2.” Please edit this sentence heavily and bring it to compliance with English grammar
We have modified our expression of this idea as shown below
Considering the kinetic diameter of the CO2 molecule is 3.3 Å (Figure 1), the passage of these molecules through the channel is impeded, due to the modifications of the channel by the high amount of water molecules, as a consequence, the adsorption of CO2 is zero.
Reviewer 2 Report
Authors have written clear manuscript. However, I have some remarks and questions to understand the manuscript better.
1) I miss the clear motivation in the manuscript for choice of this specific MOF, Mg-CUK-1 in this study. Authors have decided to keep the introduction very short and thus it lack the proper explanation with respect to this. It should be added in the manuscript.
2) The effect of confinement of water molecules on structural transformation of the MOF are clear with the study. These results clearly explain the experimental results that were obtained with the studies on this MOF in their previous work. I would like to see the experimental data of CO2 adsorption in presence and absence of water to be added in supplementary information. Without the data, the study looks incomplete.
3) There are few grammatical errors in the manuscript. Authors should proof-check it again before re-submission.
Author Response
1) I miss the clear motivation in the manuscript for choice of this specific MOF, Mg-CUK-1 in this study. Authors have decided to keep the introduction very short and thus it lack the proper explanation with respect to this. It should be added in the manuscript.
Some previous works by our research group with similar materials have been included, as well as a comparison with work with experimental and computational results, in order to give a clearer picture about the motivation of this work.
The improvement on the CO2 capture capacity of MIL-101-(Cr) has been previously demonstrated by applying a relative humidity (RH) of 10% and with a complete regeneration of the material after the adsorption process. This was confirmed experimentally and corroborated by computational simulations [6]. Very recently, thorough investigations are also being made to enhance the CO2 capture of MOFs by using the synergistic effects caused by pre-confining small amounts of polar molecules in their pores [7–9]. In this regard, we previously showed that the confinement of small amounts of H2O in a series of MOFs (functionalized with hydroxo functional groups, μ2-OH), steadily resulted in improved CO2 capture properties [10], specifically in other MOF examples constructed with this functional group (NOTT-400 [11], MIL-53-(Al) [12], InOF-1 [13], CAU-10 [14]) it has been demonstrated a strong interaction between CO2 and the hydroxo functional groups (via hydrogen bonds). Such interactions have been elegantly explained by Benoit and co-workers [9].
2) The effect of confinement of water molecules on the structural transformation of the MOF are clear with the study. These results clearly explain the experimental results that were obtained with the studies on this MOF in their previous work. I would like to see the experimental data of CO2 adsorption in presence and absence of water to be added in the supplementary information. Without the data, the study looks incomplete.
The idea of this contribution is to explain the enhancement of the CO2 capture when small amounts of water are confined within Mg-CUK-1 by experimental Le Bail refinements. Such CO2 capture enhancement has been previously reported in Ref. 17 and it was interpreted by a computation approach (Monte Carlo simulations). Thus, such CO2 capture values have now been emphasized for a better comparison in the main manuscript:
Remarkably, Mg-CUK-1, previously demonstrated a 1.8-fold increase in CO2 capture from 4.6 wt% to 8.5 wt% in presence of 0% (anhydrous conditions) and 18% of RH respectively [17], see Fig. S9.
And the experimental data of CO2 included in the MS (Fig. S9).
3) There are a few grammatical errors in the manuscript. The authors should proof-check it again before re-submission.
We have read the work and take care of the grammar details, improving this part of the content.
Reviewer 3 Report
The authors report the structural transformations of the Mg-CUK-1 metal-organic framework triggered by water adsorption, that may result in the increase of CO2 capturing. The paper can be accepted in the Materials after minor revision.
- The Introduction should be revised. Most part of this section is almost not related to the content. Instead, some phrases about stimul-responsive MOFs should be added.
- The novelty of the paper should be highlithed.
Author Response
1. The Introduction should be revised. Most part of this section is almost not related to the content. Instead, some phrases about stimul-responsive MOFs should be added.
We have corrected the work by exemplifying the use of other MOFs for the improvement of CO2 capture under conditions of relative humidity.
The improvement in the CO2 capture capacity of MIL-101-(Cr) has been previously demonstrated by applying a relative humidity (RH) of 10% and with complete regeneration of the material after the adsorption process. This was confirmed experimentally and corroborated by computational simulations [6]. Very recently, thorough investigations are also being made to enhance the CO2 capture of MOFs by using the synergistic effects caused by pre-confining small amounts of polar molecules in their pores [7–9]. In this regard, we previously showed that the confinement of small amounts of H2O in a series of MOFs (functionalized with hydroxo functional groups, μ2-OH), steadily resulted in improved CO2 capture properties [10], specifically in other MOF examples constructed with this functional group (NOTT-400 [11], MIL-53-(Al) [12], InOF-1 [13], CAU-10 [14]) it has been demonstrated a strong interaction between CO2 and the hydroxo functional groups (via hydrogen bonds). Such interactions have been elegantly explained by Benoit and co-workers [9].
2. The novelty of the paper should be highlithed.
We have added a clearer explanation highlighting the relevance and contribution of our work in the main text:
In this communication it is described, by PXRD (Le Bail methodology refinements), the structure transformation of Mg-CUK-1 (CUK for Cambridge University–KRICT, see SM, Figure S1) which is a material that contains these hydroxo functional groups μ2-OH [16], due to amount variation of confined H2O molecules within its pores, which previously demonstrated CO2 capture enhanced properties [17].